# Comparison of the 11-Day Adhesive ECG Patch Monitor and 24-h Holter Tests to Assess the Response to Antiarrhythmic Drug Therapy in Paroxysmal Atrial Fibrillation

**DOI:** 10.3390/diagnostics13193078

**Published:** 2023-09-28

**Authors:** Soohyun Kim, Young Choi, Kichang Lee, Sung-Hwan Kim, Hwajung Kim, Sanghoon Shin, Soyoon Park, Yong-Seog Oh

**Affiliations:** 1Division of Cardiology, Department of Internal Medicine, Seoul St. Mary Hospital, College of Medicine, The Catholic University of Korea, Seoul 06591, Republic of Korea; withsoohyunk@gmail.com (S.K.); sunghwan@catholic.ac.kr (S.-H.K.); hjkimmmmm@gmail.com (H.K.); to93270573@gmail.com (S.S.); syoon007@hotmail.com (S.P.); 2Cardiovascular Research Institute for Intractable Disease, College of Medicine, The Catholic University of Korea, Seoul 06591, Republic of Korea; 3Cardiovascular Research Center, Massachusetts General Hospital, Boston, MA 02114, USA; kichang.lee@mgh.harvard.edu; 4Harvard Medical School, Boston, MA 02115, USA

**Keywords:** atrial fibrillation, Holter, ambulatory electrocardiographic monitoring, antiarrhythmic drug

## Abstract

Accurate assessment of the response to the antiarrhythmic drug (AAD) in atrial fibrillation (AF) is crucial to achieve adequate rhythm control. We evaluated the effectiveness of extended cardiac monitoring using an adhesive ECG patch in the detection of drug-refractory paroxysmal AF. Patients diagnosed with paroxysmal AF and receiving AAD therapy were enrolled. The subjects simultaneously underwent 11-day adhesive ECG patch monitoring and a 24-h Holter test. The primary study outcome was a detection rate of drug-refractory AF or atrial tachycardia (AT) lasting ≥30 s. A total of 59 patients were enrolled and completed the study examinations. AF or AT was detected in 28 (47.5%) patients by an 11-day ECG patch monitor and in 8 (13.6%) patients by a 24-h Holter test (*p* < 0.001). The 11-day ECG patch monitor identified an additional 20 patients (33.8%) with drug-refractory AF not detected by the 24-h Holter, and as a result, the treatment plan was changed in 11 patients (10 catheter ablations, one medication change). In conclusion, extended cardiac rhythm monitoring using an adhesive ECG patch in patients with paroxysmal AF under AAD therapy led to over a threefold higher detection of drug-refractory AF episodes, compared to the 24-h Holter test.

## 1. Introduction

Atrial fibrillation (AF) is the most common sustained tachyarrhythmia, resulting in uncoordinated atrial electrical activation with ineffective atrial contraction [1,2]. It is associated with an increased risk of stroke, heart failure, and mortality [1,2,3], and successful rhythm control treatment of AF can result in improved clinical outcomes [4,5]. Antiarrhythmic drugs (AAD) have only modest efficacy in maintaining sinus rhythm, and catheter-based ablation therapy can be considered in AF unresponsive to AAD therapy [6,7]. However, accurate identification of AF recurrence despite pharmacologic treatment is often difficult, particularly in paroxysmal AF with low arrhythmic burden [8]. Ambulatory cardiac monitoring with a 24- or 48-h Holter test has been widely used to detect recurrent AF after treatment with AAD and/or catheter ablation, but its efficacy is yet unsatisfactory [9,10].

Recently, extended monitoring using patch-based cardiac monitoring systems has been demonstrated to have higher diagnostic yields with less patient discomfort due to wearing the device [9]. The multiday continuous cardiac monitor showed a 2–5 times higher efficacy in AF detection, compared to the conventional 24-h Holter test in patients with stroke or transient ischemic attack (TIA) [10,11,12,13,14]. Early detection of drug-refractory AF will help with appropriate decision-making for an adequate rhythm control strategy, and extended cardiac monitoring may result in the timely application of alternative treatment for rhythm control, including catheter ablation therapies. However, the benefit of long-term ECG patch monitoring for detecting drug-refractory AF has not been well demonstrated. The purpose of this study was to evaluate and compare the diagnostic accuracies of the conventional Holter test and long-term ECG patch monitoring devices in detecting recurrent paroxysmal AF that does not respond to AAD treatment.

## 2. Materials and Methods

### 2.1. Recruitment

This was a single-center prospective study. We recruited patients diagnosed with paroxysmal AF and treated with AADs for more than six weeks at our institution (Seoul St. Mary’s Hospital, Seoul, Republic of Korea) between February 2022 and June 2022. The study population included (i) adult patients (>18 years of age) with a previous documented AF on a 12-lead ECG or Holter test, (ii) patients who had treatment with class Ic or class III AADs for at least six weeks, and (iii) patients with a sinus rhythm on a 12-lead ECG at the clinical visit. Patients with the following conditions were excluded: (i) non-paroxysmal AF, (ii) hypersensitivity to components of the adhesive ECG patch monitor, or (iii) the presence of an implanted pacemaker. Of the sixty patients, one was excluded from the analysis because of death unrelated to the study procedure. The study was registered in the public clinical registry (Clinical Research Information Service [15], study no. KCT0006889) and was approved by the Institutional Review Board of the Catholic Medical Center of Korea. The study protocol conformed to the ethical guidelines of the Declaration of Helsinki. All study subjects provided written informed consent.

### 2.2. Study Protocol

A 3-channel, 24-h Holter (Evo, Spacelabs Healthcare, Snoqualmie, WA, USA) and a long-term adhesive ECG patch monitoring device (AT-Patch^®^, ATsens Co., Ltd., Seongnam-si, Republic of Korea) were applied simultaneously to each patient (Figure 1). The AT-Patch^®^ is an adhesive, removable, and waterproof single-lead ECG monitoring device capable of recording up to 11 consecutive days. It weighs about 13 g and measures 93.0 × 50.6 × 8.3 mm. The study coordinator applied the device to the left pectoral region of the patient’s chest in the second intercostal space towards the heart. AAD therapy was maintained throughout the study period. The type and dosage of medication were determined based on the physician’s discretion. Patients were scheduled to visit the clinic to remove the Holter on day one and the patch devices on day twelve. The twenty-four-hour Holter and ECG patch monitor data were forwarded to the data analysis laboratory. When the patient visited the clinic to remove the device, side effects related to the attached device and the patient’s symptoms were examined. 

### 2.3. Outcome Definition

The primary study outcome was the detection rate of AF or atrial tachycardia (AT) lasting longer than 30 s. Secondary outcomes were detection rates of ventricular
tachycardia (≥3 consecutive beats of wide QRS tachycardia), sinus bradycardia (<40 beats/minute), pause (>3 s), and 2nd- or 3rd-degree atrioventricular (AV) block. The diagnosis of AF was defined according to the previous guideline, which includes the electrocardiographic features of irregularly irregular RR intervals and the absence of distinct, repeating *p*-waves [6]. An AT was diagnosed as a narrow QRS tachycardia with discrete repetitive *p*-waves that had a different morphology, compared to those during the sinus rhythm. For the interpretation of long-term patch ECG data, we used a unique arrhythmia-specific screening algorithm. After initial signal filtering and noise canceling, AF episodes were automatically screened based on the variability and complexity of the RR interval. The algorithm also facilitated the interpretation of the secondary outcomes by screening and providing episode strips of wide QRS tachycardias, bradycardias, irregular RR intervals apart from AF, or changes in QRS morphology according to the prespecified threshold. After initial algorithm screening, provided event strips were reviewed and analyzed by trained electrophysiologists. For the 24 hr Holter test, all electrocardiographic data were analyzed manually. Two electrophysiologists independently performed the electrogram review process, and the diagnosis of each arrhythmia was confirmed when both opinions were consistent. After completing the study examinations, patients were followed up for 6 months, and any changes in AF treatment regarding rhythm control were analyzed. These treatment changes included a catheter ablation, alterations in AAD type or dosage, or discontinuation of AAD.

### 2.4. Statistical Analysis

For the sample size calculation, the detection rate of drug-refractory AF of the 11-day monitor was estimated to be 48% [2], which was expected to be 50% higher than the 24-h Holter [12]. It was assumed that 4% of patients with a positive Holter test would be negative on the ECG patch monitor. An overall sample size of 60 was calculated to have 80% power to detect a statistical difference between the two diagnostic modalities at a two-tailed alpha of 0.05, considering a 5% dropout rate.

Categorical variables were presented as numbers and percentages and continuous variables as the mean ± standard deviation. The arrhythmia detection rates, including the primary outcome (AF/AT detection rate) and the secondary outcomes, were pairwise compared between the 24-h Holter and the ECG patch monitor using McNemar’s test. Non-paired categorical variables were compared using the chi-square test or Fisher’s exact test, as appropriate. All analyses were two-sided, and the statistical significance was considered when a *p*-value was less than 0.05. All statistical analyses were performed using R version 4.2.1 (R Foundation).

## 3. Results

### 3.1. Baseline Characteristics

A total of 59 patients received the ECG patch monitor and Holter and were included in the study analysis. The baseline characteristics of the enrolled patients are summarized in Table 1. The mean age of the enrolled patients was 67.0 ± 10.5 years, and 31 (52.5%) were female. The median time interval between the initial AF diagnosis and study inclusion was 34.2 (15–74) months. The average time of wearing the ECG patch monitor was 10.6 days. Fifty-seven patients completed ECG patch monitoring for >8 days, and 54 completed continuous ECG patch monitoring for 11 days. The type of AAD used during the study examination was mostly pilsicainide (72.9%), followed by propafenone and dronedarone.

### 3.2. Detection Rates of Arrhythmias

The long-term ECG patch monitor detected AF in 28 (47.5%) patients. AT was detected by the ECG patch monitor in four (6.8%) patients, and all were accompanied by AF. The Holter test detected four (6.8%) AF patients and four (6.8%) AT patients. All four AF events on the Holter were also detected as AF on the long-term patch monitor. Two of Holter’s diagnosed AT events were interpreted as AF on the long-term patch monitor (Figure 2), and the other two AT events were also interpreted as AT on the patch monitor. The overall detection rate of AF or AT was 3.5-fold higher on the long-term patch monitor, compared to the 24-h Holter (47.5% vs. 13.6%, respectively, *p* < 0.001) (Table 2). The prevalence of other arrhythmias, including ventricular tachyarrhythmias and bradyarrhythmia, was low, and there was no significant difference in the detection rates between the patch monitor and the Holter test (Table 3). Detection rates of clinically significant sinus bradycardia (<40 beats/min) or sinus pause (>3 s) were numerically higher on the long-term patch monitor (6.8% vs. 3.4%, *p* = NS), and a second-degree AV block was detected in the two (3.4%) same patients on the ECG patch monitor and the Holter test. During the study period, 29 (49.2%) patients reported AF-related symptoms, such as palpitations or the sensation of irregular heartbeats. AF/AT detection rates on the ECG patch monitor did not differ between the symptomatic (15/29, 51.7%) and asymptomatic (13/30, 43.3%) patients (*p* = 0.701). The efficacy of the ECG patch monitor in detecting AT/AF was 5-fold higher compared to the Holter test in symptomatic patients (51.7% vs. 10.3% in the patch monitor and Holter test, respectively) and was 2.6-fold higher in asymptomatic patients (43.3% vs. 16.6%).

### 3.3. Temporal Trends in AF Detection on the ECG Patch Monitor

A total of 87 AF episodes were detected by the ECG patch monitor in 28 patients, and the average number of days with AF episodes per patient was 3.1 days. AF/AT was detected within the first 48 h in 15 (53.6%) patients, and most AF/AT (23/28, 82.1%) were identified within 96 h (Figure 3). In addition, the ECG patch monitor identified two more patients with drug-refractory AF/AT within 7 days and another three patients on days 8 and 11.

### 3.4. Treatment after AF Detection

Out of 28 patients with drug-refractory AF, 18 underwent catheter ablation for AF, and 1 experienced a modification in AAD therapy, which was specifically a dose escalation of pilsicainide, in the 6 months following the study examinations. All eight patients who had drug-refractory AF or AT documented by both the Holter and ECG patch monitor underwent catheter ablation. Among the 20 patients with drug-refractory AF detected solely by the ECG patch monitor, the rhythm control treatment was modified in 11 (55%) patients: 10 underwent catheter ablation, and 1 had a change in AAD dose (as described above).

### 3.5. Adverse Events

After removing the ECG patch monitor, five patients showed adverse skin reactions. Three patients showed skin pruritis, and two reported minor vesicles around the patch. Two patients had their patch devices removed two days early due to skin discomfort. There were no serious adverse reactions requiring medical therapy, and all skin lesions were self-limited within a few days. There were no skin lesions or patient complaints related to the 24-h Holter device.

## 4. Discussion

### 4.1. Summary of the Results

This study compared the efficacy of a patch-based single-lead ECG monitoring device for 11 days with a 24-h Holter system in detecting drug-refractory paroxysmal AF. The main findings of this study were that (i) extended monitoring with an ECG patch device increased the AF detection rate more than three-fold, (ii) the efficacy of the single-lead ECG patch device was consistently higher in patients with or without relevant symptoms, and (iii) most (82.1%) of the patients with drug-refractory AFs were detected within 96 h on the ECG patch monitor. However, continuous monitoring over 11 days consistently detected additional cases of drug-refractory AF thereafter. Additionally, (iv) extended monitoring using an ECG patch monitor increased the detection of drug-refractory AF and resulted in treatment plan changes in an additional 11 (18.6%) patients.

### 4.2. Detection for Drug-Refractory AF by Extended Monitoring

Various cardiac rhythm monitoring systems are currently available, including Holter tests, multiday continuous ECG monitors, implantable cardiac monitors, and watch-based devices [16,17,18,19,20,21]. A patch-based adhesive monitoring device is useful for obtaining stable long-term cardiac rhythm data and detecting undiagnosed AF in a non-invasive manner [17]. Up to two weeks of cardiac monitoring using an adhesive patch monitoring device resulted in a 4–10 times higher AF diagnosis rate than the usual monitoring strategy in patients without a known history of AF [17,22]. After AF is diagnosed, appropriate assessment of the rhythm control status is also essential in that long-term maintenance of the sinus rhythm can benefit clinical outcomes [5,23]. Although various AADs are available for AF rhythm control treatment, the overall AF recurrence rate in patients receiving AAD has been reported to be 35–65% [24,25,26]. In this study, the detection rate of the AF refractory to AAD was 47.5% on the ECG patch monitor, consistent with previous data. The AF detection rate in the 24-h Holter test was only 13.6%. The status of AF rhythm control has been commonly assessed by a 24 to 48 hr Holter in previous studies [7,27], but our study result implies that this approach can significantly underestimate AF recurrence in this patient group. In addition, it is inferred that long-term ECG monitoring should be considered even in asymptomatic patients receiving drug therapy, considering that recurrent AF was detected in 43.3% of patients who did not report AF-related symptoms in this study.

### 4.3. Advantage and Limitation of the Adhesive Patch Monitor

Prolonged ECG recordings using an intermittent patient-triggered monitor or telemetry device can also improve the detection efficacy of AF [28]. However, previous studies have reported substantially reduced adherence to extended monitoring using multi-lead systems or event monitors, which can be attributed to significant skin irritation, device bulkiness, and interference with patient’s work and public activities [29,30,31]. The patch-based single-lead ECG monitor has several advantages, compared to other extended cardiac monitoring devices; it is relatively small in size, water resistant, capable of long-term adhesion, and easier to wear, due to the leadless system [32]. Despite the advantages of the patch monitor, concerns still remain about the electrogram resolution and diagnostic accuracy of a single-lead system, such as misreading aberrant conduction due to low *p*-wave amplitudes and under-sensing low-amplitude, non-conducting premature atrial complexes [33,34]. Nevertheless, the overall AF detection accuracy for single-lead ECG has been reported to be satisfactory and may be further improved using an automated detection algorithm [35]. This issue is particularly important in patients with advanced atrial remodeling and can result in difficulty identifying any atrial electrical activity [36]. In our study, two of the four AT events detected by the 24-h Holter were interpreted as AF by the single-lead ECG patch monitor. This discrepancy resulted from the low resolution of the ECG patch monitor for identifying regular *p*-waves in AT, due to the spatial limitation of the short patch device. Previous studies also reported only moderate accuracy in atrial flutter diagnosis by the patch monitor [34]. Proper device placement and technical improvement would overcome this shortcoming of a small single-lead ECG [37].

### 4.4. Clinical Implication

Current guidelines recommend the rhythm control strategy for AF to relieve symptoms and improve quality of life, and catheter ablation is effective in patients who are intolerant or refractory to AADs [6]. Additionally, a recent study reported that early rhythm control therapy resulted in better clinical outcomes than conventional care, defined as a composite of cardiovascular death, stroke, or hospitalization due to heart failure or acute coronary syndrome [4]. For effective rhythm control, it is crucial not only to select the rhythm control strategy in the early period after AF diagnosis but also to appropriately assess the maintenance of the sinus rhythm. In our study, an 11-day ECG monitor could identify 20 (32.8%) patients with drug-refractory AF in whom a 24-h Holter test did not detect AF. The timely detection of unsuccessful AF control led to the decision for catheter ablation in 10 (16.9%) patients. AF was detected within four days in most cases, but 18% of drug-refractory AF was first detected between days 5 and 11. The findings suggest that the optimal monitoring duration for enhanced AF detection would not be shorter than our study period. Current adhesive patches, including the latest version of the AT patch, offer noninvasive cardiac monitoring for up to 14 days, and this approach can be suggested for patients receiving AAD therapy for AF [22]. In our data, nearly half of the enrolled patients were asymptomatic, but the prevalence of drug-refractory AF was not significantly different between the symptomatic and asymptomatic patients. The advantage of the long-term patch monitor compared to the 24-h Holter in the diagnosis of AF appears less pronounced in asymptomatic patients than in symptomatic patients; nevertheless, the diagnostic yield was approximately 2.6 times higher in the patch monitor. The primary indication of catheter ablation is for the symptomatic control for AF [6]; however, a strategy of rhythm control can improve clinical outcomes in asymptomatic patients, particularly in those with paroxysmal AF [38]. Although it remains unclear whether indiscriminate screening of cardiac rhythm status benefits asymptomatic patients with paroxysmal AF, the adhesive patch monitor can provide information regarding the appropriateness of current therapeutic interventions while minimizing patient discomfort during the examination. Further research is warranted to assess the potential benefit on clinical outcomes of extended cardiac monitoring in AF patients on AAD therapy.

### 4.5. Limitations

Our study had some limitations. This was a single-center study with a relatively small sample size to compare the accurate diagnosis performances of the ECG patch monitoring device and the Holter. Nevertheless, enrolled patients wore both devices simultaneously to compare the effectiveness of each device directly, and the study results were sufficient to demonstrate a clear benefit of the ECG patch monitor. Five (8.5%) patients who did not complete the 11-day ECG patch monitoring were included in the study analyses, which could have contributed to the underestimation of the detection rate of the ECG patch monitor. However, the average patch-wearing period was sufficient (10.6 days), and the AF detection rate of the ECG patch monitoring device was sufficiently higher than that of the Holter, so a clear conclusion could be drawn. Third, the types and doses of prescribed AADs were not strictly defined. Fourth, we did not necessarily perform echocardiography for study inclusion, and detailed echocardiographic data on all patients were not available. Finally, we analyzed the data regarding clinical treatment changes, but we could not compare the outcomes associated with the two modalities, since the patients were not grouped according to the study examination.

## 5. Conclusions

In our prospective study, 11-day continuous monitoring using a single-lead adhesive ECG patch was superior to a conventional 24-h Holter in detecting AAD-refractory AF or AT. The usefulness of the continuous patch monitor was demonstrated in both patients presenting with AF-related symptoms and those without. The extended cardiac rhythm monitor provides higher diagnostic results for paroxysmal arrhythmias and would help determine the appropriate therapeutic decision for patients with AF who have already been diagnosed and treated.

## Figures and Tables

**Figure 1 diagnostics-13-03078-f001:**
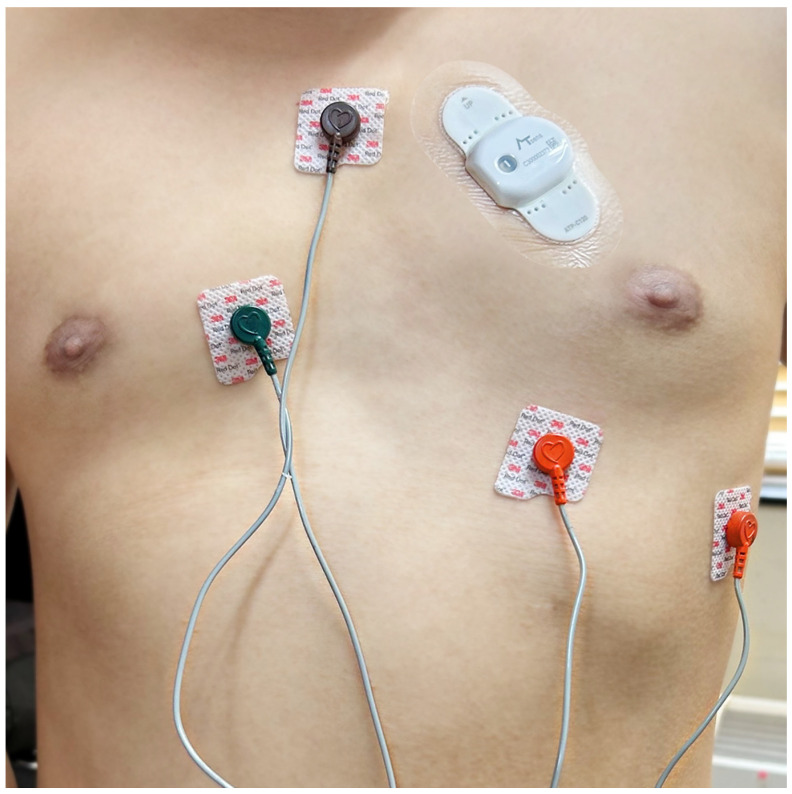
An adhesive ECG patch monitor and a Holter monitor are simultaneously attached to the patient’s chest.

**Figure 2 diagnostics-13-03078-f002:**
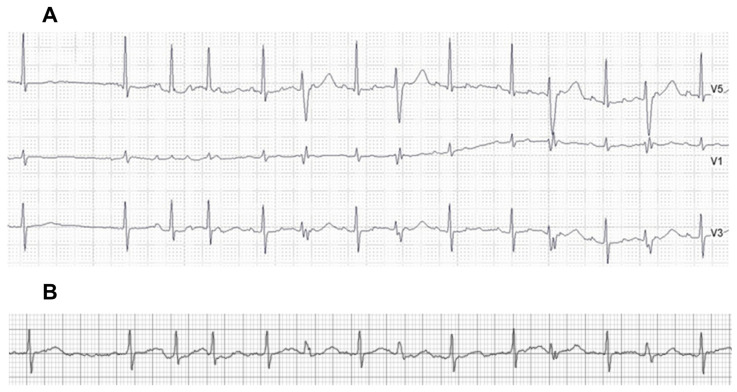
AT with discernable *p*-waves on the Holter test (**A**), which was interpreted as AF on the ECG patch monitor on the same day (**B**). AT = atrial tachycardia; AF = atrial fibrillation.

**Figure 3 diagnostics-13-03078-f003:**
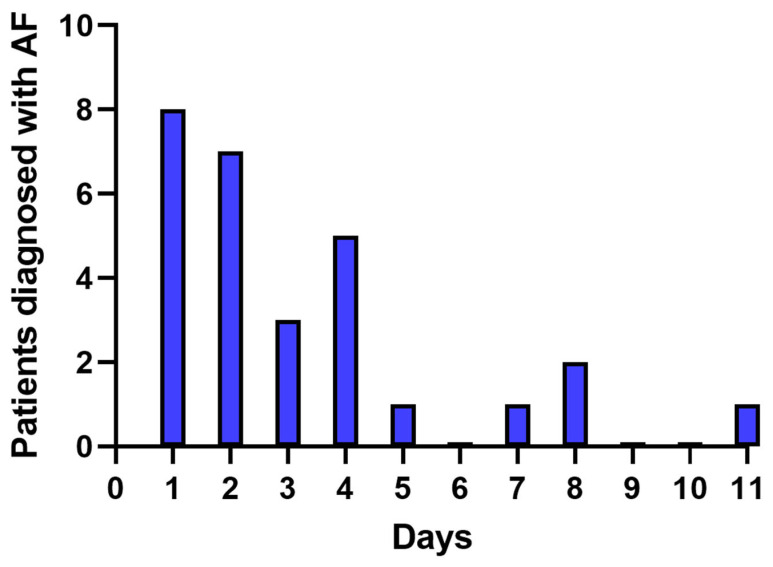
The number of patients with AF (or AT) detection by the ECG patch monitor per day. AF = atrial fibrillation; AT = atrial tachycardia.

**Table 1 diagnostics-13-03078-t001:** Baseline characteristics of the patients.

Variables	Total *N* = 59
Age, years	67.0 ± 10.5
Female, *n* (%)	31 (52.5%)
Heart rate, beats/min	71 ± 14
CHA_2_DS_2_-VASc score	2.8 ± 1.5
AF duration, months	34.2 (15–74)
Patch wearing time, days	10.6 ± 1.48
Comorbidities, *n* (%)	
Hypertension	51 (86.4%)
Diabetes	11 (18.6%)
Prior stroke	26 (44.1%)
Dyslipidemia	44 (74.6%)
Heart failure	8 (13.6%)
Coronary artery disease	2 (3.4%)
Type of anticoagulant, *n* (%)	
DOAC	44 (74.6%)
Warfarin	1 (1.7%)
Antiplatelet	3 (5.1%)
Type of antiarrhythmic drugs, *n* (%)	
Propafenone	7 (11.9%)
Flecainide	3 (5.1%)
Pilsicainide	43 (72.9%)
Dronedarone	4 (6.8%)
Other medications	
ACEi/ARB	49 (83.1%)
Beta Blocker	49 (83.1%)
Diuretics	5 (8.5%)
CCB	39 (66.1%)

Categorical variables are presented as numbers (percentages), and continuous variables are presented as the mean ± standard deviation. AF = atrial fibrillation, DOAC = direct oral anticoagulants, ACEi = angiotensin-converting enzyme inhibitor, ARB = angiotensin receptor blocker, CCB = calcium channel blocker.

**Table 2 diagnostics-13-03078-t002:** The number of patients with AF (or AT) detection by 24-h Holter and the patch monitor.

		Holter	
		No	Yes
Patch ECG monitor	No	31	0
	Yes	20	8

**Table 3 diagnostics-13-03078-t003:** Arrhythmias detected on the 11-day adhesive ECG patch monitor and the 24-h Holter.

Arrhythmias, *n* (%)	Patch Monitor	Holter
AF or AT	28 (47.5%)	8 (13.6%)
AF	28 (47.5%)	4 (6.8%)
AT	4 (6.8%)	4 (6.8%)
Non-sustained VT	1 (1.7%)	0
Sinus node dysfunction *	4 (6.8%)	2 (3.4%)
2nd-degree AV block	2 (3.4%)	2 (3.4%)

* Sinus bradycardia with heart rate less than 40 beats/min or sinus pause >3 s. AF = atrial fibrillation, AT = atrial tachycardia, VT = ventricular tachycardia, AV = atrioventricular.

## Data Availability

The datasets analyzed in this study are not publicly available but are available from the corresponding author upon reasonable request.

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
