# Peer review of "Comparison of the 11-Day Adhesive ECG Patch Monitor and 24-h Holter Tests to Assess the Response to Antiarrhythmic Drug Therapy in Paroxysmal Atrial Fibrillation"

_diagnostics, 2023, doi:10.3390/diagnostics13193078_

Round 1

Reviewer 1 Report

The paper evaluated the effectiveness of extended cardiac monitoring using an adhesive ECG patch in detecting drug-refractory paroxysmal atrial fibrillation. The subjects simultaneously underwent 11-day adhesive ECG patch monitoring and a 24-hour Holter test. Fifty-nine patients diagnosed with paroxysmal atrial fibrillation and receiving antiarrhythmic drug therapy were enrolled.

The study protocol conforms to the ethical guidelines of the Declaration of Helsinki. The study was registered in the public clinical registry (Clinical Research Information Service, study no. KCT0006889) and was approved by the Institutional Review Board of the Catholic Medical Center of Korea.

All study subjects provided written informed consent.

The study methods are appropriate, and the data are valid. The results are well highlighted, and the conclusions are adequate.

The references are relevant and correctly chosen, and related work is discussed and cited appropriately.

Minor comments:

In the ”Materials and Methods” Section, the authors should explain why they chose 11 days for ECG Patch monitoring.

Author Response

We responded to the comments below and indicate how the manuscript has been modified to respond to these comments. Revisions are indicated in red font in the main text.

Author’s response: We appreciate your generous comments. For your minor comment, we sought to monitor over the longest possible period using the patch device. At the time of the study, the adhesive patch used (AT patch) was capable of monitoring for a maximum of 11 days, so we chose 11 days for the study procedure. After the completion of the study, the device has been updated, and it is now capable of recording up to 14 days. We changed the original manuscript and added it in discussion.

Manuscript change: Page 8, discussion

“The findings suggest that the optimal monitoring duration for enhanced AF detection would not be shorter than our study period. Current adhesive patches, including the latest version of the AT-patch offer noninvasive cardiac monitoring for up to 14 days, and this approach can be suggested for patients receiving AAD therapy for AF [22].”

Reviewer 2 Report

1. The "p" used to indicate test differences should be italicized.

2. What are the methods for statistical analysis? Can you provide a detailed explanation?

3.In this study, we can see that AT and AF are important features. How did this study differentiate these features? Additionally, the chapter lacks detailed explanations for the definitions of many features in relation to the outcome definition.

4.While this study has identified some observed features, they have all been manually labeled. Are there any other, more scientific or mathematical approaches to identifying corresponding features?

Author Response

We very much appreciate the careful review and helpful comments. We responded to the comments below and indicate how the manuscript has been modified to respond to these comments. Revisions are indicated in red font in the main text.

  1. The "p" used to indicate test differences should be italicized.

We changed “p” into italicized letter throughout the manscript.

  1. What are the methods for statistical analysis? Can you provide a detailed explanation?

Author’s response: Because the arrhythmia detection rate data was paired for each patient between the 24-hr Holter and patch monitor, we used McNemar’s test to compare the diagnostic performance of the two modalities. Additionally, chi-square test was used for the comparison of non-paired categorical variable comparison (e.g. comparison between the symptomatic and asymptomatic patients)

Manuscript change: (page 4, Method)

“The arrhythmia detection rates including the primary outcome (AF/AT detection rate) and the secondary outcomes were pairwise compared between the 24-hour Holter and the ECG patch monitor using McNemar’s test. Non-paired categorical variables were compared using the chi-square test or Fisher’s exact test, as appropriate”

  1. In this study, we can see that AT and AF are important features. How did this study differentiate these features? Additionally, the chapter lacks detailed explanations for the definitions of many features in relation to the outcome definition.

Author’s response: We added more detailed information about how we defined the AT and AF in this study. For the secondary outcomes, detailed diagnostic criteria for each arrhythmia were not elaborated upon, given that they are medically well-established. Instead, a brief definition was provided in parentheses next to each arrhythmia.

Manuscript change: (Page 2, Method – outcome definition)

“Secondary outcomes were detection rates of ventricular tachycardia (≥3 consecutive beats of wide QRS tachycardia), sinus bradycardia (<40 beats/minute), pause (>3 seconds), and 2nd or 3rd-degree atrioventricular (AV) block. The diagnosis of AF was defined according to the previous guideline, that includes the electrocardiographic features of irregularly irregular R-R intervals and the absence of distinct repeating P-waves [6]. An AT was diagnosed as a narrow QRS tachycardia with discrete repetitive P-waves that had a different morphology compared to those during the sinus rhythm.”

4.While this study has identified some observed features, they have all been manually labeled. Are there any other, more scientific or mathematical approaches to identifying corresponding features?

Author’s response: For the interpretation of the long-term patch ECG data, we used automatic rhythm analyzing algorithm for initial screening of the event episodes. We added these explanations in the method section.

Manuscript change: (Page 3, method- outcome definition)

 “For the interpretation of long-term patch ECG data, we used unique arrhythmia-specific screening algorithm. After initial signal filtering and noise cancelling, AF episodes were automatically screened based on the variability and complexity of RR interval. The algorithm also facilitated the interpretation of the secondary outcomes by screening and providing episode strips of wide QRS tachycardias, bradycardia, irregular RR intervals apart from AF or changes in QRS morphology according to the prespecified threshold. After initial algorithm screening, provided event strips were reviewed and analyzed by trained electrophysiologists. For the 24-hr Holter test, all electrocardiographic data was analyzed manually. Two electrophysiologists independently performed the electrogram review process, and the diagnosis of each arrhythmia was confirmed when both opinions were consistent.”

Round 2

Reviewer 2 Report

1. Authors revised the draft according to the reviewer's comments.